# Poor Glycemic Control Can Increase the Plasma Kidney Injury Molecule-1 Concentration in Normoalbuminuric Children and Adolescents with Diabetes Mellitus

**DOI:** 10.3390/children8050417

**Published:** 2021-05-19

**Authors:** Moon Bae Ahn, Kyoung Soon Cho, Seul Ki Kim, Shin Hee Kim, Won Kyoung Cho, Min Ho Jung, Jin-Soon Suh, Byung-Kyu Suh

**Affiliations:** 1Department of Pediatrics, Seoul St. Mary’s Hospital, College of Medicine, The Catholic University of Korea, Seoul 06591, Korea; mbahn@catholic.ac.kr; 2Department of Pediatrics, Bucheon St. Mary’s Hospital, College of Medicine, The Catholic University of Korea, Bucheon 14647, Korea; soon926@hanmail.net; 3Department of Pediatrics, Eunpyeong St. Mary’s Hospital, College of Medicine, The Catholic University of Korea, Seoul 03312, Korea; seulki12633@gmail.com; 4Department of Pediatrics, Incheon St. Mary’s Hospital, College of Medicine, The Catholic University of Korea, Incheon 21431, Korea; tigger1018@naver.com; 5Department of Pediatrics, St. Vincent hospital, College of Medicine, The Catholic University of Korea, Suwon 16247, Korea; wendy626@catholic.ac.kr; 6Department of Pediatrics, Yeouido St. Mary’s Hospital, College of Medicine, The Catholic University of Korea, Seoul 07345, Korea; jmhpe@catholic.ac.kr

**Keywords:** kidney injury molecule-1, diabetic nephropathy, diabetes mellitus, chronic kidney disease

## Abstract

Diabetic nephropathy (DN) is a serious microvascular complication in childhood diabetes and microalbuminuria has been a solid indicator in the assessment of DN. Nevertheless, renal injury may still occur in the presence of normoalbuminuria (NA) and various tubular injury biomarkers have been proposed to assess such damage. This case-controlled study aimed to evaluate plasma and urinary neutrophil gelatinase-associated lipocalin and kidney injury molecule-1 (KIM-1) levels in diabetic children particularly in those with normo- and high-NA stages and determine their role in predicting DN. Fifty-four children/adolescents with type 1 and 2 diabetes and forty-four controls aged 7–18 years were included. The baseline clinical and laboratory characteristics including plasma and urinary biomarkers were compared. The plasma KIM-1 levels were significantly higher in diabetic children than in the controls and in high-NA children than normo-NA children. Glycosylated hemoglobin (HbA1c) was identified as a significant risk factor for increased plasma KIM-1. The optimal cutoff for HbA1c when the plasma KIM-1 was > 23.10 pg/mL was 6.75% with an area under the curve of 0.77. For diabetic children with mildly increased albuminuria, the plasma KIM-1 complementary to MA may help increase the yield of detecting DN. Our findings also suggested an HbA1c cutoff of 6.75% correlated with increased plasma KIM-1.

## 1. Introduction

Diabetic nephropathy (DN) is considered to be one of the most serious microvascular complications in terms of the mortality and morbidity of diabetic patients. The blood urea nitrogen level, creatinine (Cr) level, estimated glomerular filtration rate (eGFR) and urine protein and albumin levels are clinical determinants of the presence and progression of DN. Despite a poor glycemic control, the risk of developing clinically evident DN during the earlier course of childhood diabetes is not commonly confronted [1]. Nevertheless, early screening should not be overlooked as chronic kidney disease (CKD) with normoalbuminuria (NA) has become a prevalent variant of DN [2]. Albuminuria is a well-recognized and early marker not only of DN but also of the progression of vascular complications such as cardiovascular diseases [3,4].

Microalbuminuria (MA) has been utilized as an indicator of incipient DN and the severity of DN can depend upon the degree of MA [5]. Nevertheless, numerous studies have demonstrated that MA could not be a specific indicator of renal injury and its utility in predicting DN progression is limited because various pathological changes and an eGFR decline are observed in the presence of NA [2,6]. Progressive renal decline could be initiated at even 10% NA, 30% MA and 50% proteinuric states in DM patients [7]. Other vascular complications such as diabetic retinopathy and cardiovascular diseases may occur in diabetic patients in a normal to mildly increased albuminuric state; thus, recent studies have suggested a revised threshold for MA [8,9].

The pathophysiology of DN includes multifactorial interactions between glucose-dependent pathways and various vasoactive hormones generating reactive oxygen species that damage not only the glomerulus but also the podocytes and tubulointerstitium [10]. The renal tubulointerstitium plays an integral role in the pathogenesis of DN and is associated with the progressive decline of renal function [11]. As albuminuria is a consequence of hypoxic tubulointerstitial damage leading to the impaired reabsorption of tubular albumin, various tubular injury biomarkers have been proposed to assess such damage [12,13]. A few biomarkers associated with tubular injury have exhibited their ability to detect early renal damage and predict progressive renal decline [14].

Neutrophil gelatinase-associated lipocalin (NGLA) is a 25 kDa protein first purified from a human neutrophil of the innate immune system in the early 1990s that is upregulated in response to kidney injury and involved in the post-injury formation and repair of the tubular epithelium [15]. Both plasma and urinary NGAL are known as useful markers and predictors of CKD progression and can be used in the early diagnosis of DN. A recent meta-analysis also suggested that the diagnostic value of NGAL was superior to other renal injury markers in patients with both microalbuminuria and macroalbuminuria [16,17]. Kidney injury molecule-1 (KIM-1), also known as Hepatitis A Virus Cellular Receptor 1 or T cell immunoglobulin mucin 1, is a type 1 transmembrane glycoprotein present on the apical membrane of proximal tubular cells and since its discovery in the late 1990s it was found to be upregulated in response to tubular damage [15,18]. KIM-1 is not detected when the kidney function is normal but is markedly increased upon tubulointerstitial damage [11,19]. According to a systemic review by Kapoula et al., urine KIM-1 could be considered to be a valuable biomarker for the early detection of DN in patients with type 2 diabetes mellitus (T2DM) [20]. Nowak et al. reported that plasma KIM-1 was strongly associated with the risk of early progressive renal decline regardless of the baseline clinical characteristics such as proteinuria or albuminuria [7].

However, clinical evidence is limited regarding the value of NGAL and KIM-1 in diabetic children and adolescents and also normal healthy children. Therefore, the aim of the present study was to investigate the plasma and urinary NGAL and KIM-1 levels of diabetic children particularly in those with a normo- and high-NA stage and determine their role in predicting renal injury. Additionally, we explored the association of the clinical and metabolic parameters of diabetic children with NGAL and KIM-1 to suggest an optimal glycemic target for the possible prevention of early-stage DN.

## 2. Methods

### 2.1. Study Participants

This case-controlled study included 54 children and adolescents diagnosed with DM, either type 1 diabetes mellitus (T1DM) or T2DM and 44 controls aged 7–18 years who visited Seoul and Bucheon St. Mary’s hospital, Korea, between July and December 2020. All T1DM children were treated with a subcutaneous insulin injection whereas T2DM children were treated with either an oral hypoglycemic agent (metformin) or lifestyle modification. Patients with other types of DM such as monogenic- or glucocorticoid-induced DM were excluded. Patients with acute kidney injury or end-stage renal disease (ESRD) who were receiving renal replacement therapy were also excluded. The controls included those who visited the outpatient clinic for growth assessment and were recruited in the absence of any underlying infectious, endocrine, neoplastic or psychiatric disorders.

This study was approved by the Institutional Review Board of the Catholic University of Korea, Seoul St. Mary’s Hospital (XC20EIDI0006) and conducted in accordance with the guidelines of the Declaration of Helsinki. All participants and their guardians provided written informed consent.

### 2.2. Definition of Diagnosing DM

The diagnosis of DM was confirmed when the following criteria were met in the presence of the classic symptoms of hyperglycemia (polyuria, polydipsia, polyphagia or weight loss): (i) random blood glucose levels of ≥ 200 mg/dL, (ii) 8 h fasting blood glucose levels of ≥ 126 mg/dL, (iii) 2 h post-load glucose levels of ≥ 200 mg/dL or (iv) repeated glycosylated hemoglobin (HbA1c) levels of ≥ 6.5% [21]. Among those with DM, T1DM was confirmed in the presence of at least one anti-pancreatic autoantibody (glutamic acid decarboxylase 65 autoantibodies, tyrosine phosphatase-like insulinoma antigen 2, insulin autoantibodies or β-cell-specific zinc transporter 8 autoantibodies). Those who did not meet the criteria of T1DM were classified as T2DM.

### 2.3. Clinical and Laboratory Data

The height (Harpenden Stadiometer, Holtain^®^, Crymych, UK) and weight (Simple Weighing Scale, Cas^®^, Korea) were recorded at the time of the sample collection and the body mass index (BMI) was calculated (kg/m^2^). All anthropometric data were converted to age- and sex-matched standard deviation scores (SDSs) using the national growth chart [22].

Fasting blood and spot urine samples were collected from participants during their visit to the outpatient clinic. The blood analysis included serum hemoglobin, C-reactive protein (CRP), glucose, Cr, protein, albumin, aspartate aminotransferase (AST), alanine aminotransferase (ALT), total cholesterol (TC), triglyceride (TG), high-density lipoprotein cholesterol (HDL-C), low-density lipoprotein cholesterol (LDL-C), HbA1c, c-peptide, insulin and uric acid. A urine analysis included protein, albumin and Cr. A urine sediment analysis was also performed to exclude urinary tract infections.

The eGFR was calculated to measure the kidney function using the revised Schwartz formula: 0.413 × height (in cm)/serum creatinine (mg/dL) [23]. The homeostasis model assessment of insulin resistance (HOMA-IR) and β-cell function (HOMA-β) indices were calculated based on the following formula: HOMA-IR = insulin (mU/L) × glucose (mg/dL)/405 while HOMA − β = (360 × fasting insulin (mU/L))/(glucose − 63) [24].

The urinary protein and albumin concentrations were adjusted to the urinary Cr concentration as a urinary protein/Cr ratio (UPCR) and an albumin/Cr ratio (UACR). Diabetic children were then subcategorized into three groups based on the degree of the UACR: normo-NA (UACR of < 10 mg/g Cr), high-NA (UACR of 10–30 mg/g Cr) and MA (UACR of > 30 mg/g Cr) to observe the differences in the levels of plasma and urinary NGAL and KIM-1 [25].

### 2.4. Measurements of Renal Injury Markers

Upon collection, blood and urine samples were immediately (within 30 min) centrifuged at 3000× *g* at 4 °C for 15 min (U-32012 Centrifuge, Boeco^®^, Hamburg, Germany) and stored at −80 °C until assayed. The concentrations of plasma and urinary NGAL and KIM-1 were determined using commercially available enzyme-linked immunosorbent assays (NGAL, Human Lipocalin-2/NGAL Immunoassay, R&D Systems^®^, Minneapolis, MN, USA; KIM-1, Human TIM-1/KIM-1/HAVCR Immunoassay, R&D Systems^®^, USA) in accordance with the manufacturer’s instructions and the pre-test recommended dilution factor provided by the individual assay kits. A microplate reader (SpectraMax 190 and SoftMax Pro7.0.2, Molecular Devices^®^, San Jose, USA) was used for the determination of concentrations and all samples were run in duplicate. The average coefficients of variation for intra-assay (inter-assay) precisions were 4.0% (6.7%), 4.0% (6.7%), 2.9% (6.4%) and 3.0% (6.2%) for plasma NGAL, urinary NGAL, plasma KIM-1 and urinary KIM-1, respectively. A few of the urinary NGAL and KIM-1 concentrations were under the detectable range and were therefore approximated to the lowest measurement of the population. Although not completely within the standard range, several concentrations were predicted based on the trend of the standard curve. The urinary levels of NGAL and KIM-1 were adjusted to the urinary Cr concentration.

### 2.5. Statistical Analysis

The statistical analysis included the descriptive analysis of all variables expressed as the median (interquartile range 25–75%) after checking the distribution normality using the Shapiro–Wilk test. The categorical variables (sex) were expressed as frequencies. The comparisons of the two groups (controls vs. diabetic children) were carried out using the Mann–Whitney U test while the comparisons of the three subcategorized groups based on MA status were carried out using the Kruskal–Wallis test. A Spearman’s rank correlation was used to study the correlations between the clinical data and renal injury markers whereas univariate and multi-variate regression analyses were used to estimate the beta coefficients (OR) and 95% confidence intervals (CI) for the increase in the levels of renal injury markers. A receiver operating characteristic (ROC) curve was generated and the area under the curve (AUC) was calculated to identify the optimal glycemic target. All statistical analyses were performed using SPSS version 24.0 (IBM Corp^®^, Armonk, NY, USA).

## 3. Results

### 3.1. Baseline Characteristics and Biomarker Measurements

This case-controlled study included 54 children with DM and 44 controls with a median age of 12.42 years (9.56–16.17 years). Among those with DM, the number of children with T1DM was 31 (57.4%) and T2DM was 23 (42.6%). All children with T1DM were treated with a subcutaneous insulin injection while 19 (82.6%) children with T2DM were treated with metformin. The number of children with MA was 10 (18.5%) among diabetic children while 1 (2.3%) child from the control group had MA. The baseline clinical and laboratory characteristics including plasma and urinary biomarker measurements in the diabetic children and controls are presented in Table 1. Age (*p* < 0.001), serum CRP (*p* < 0.001), glucose (*p* < 0.001), Cr (*p* < 0.001), HbA1c (*p* < 0.001) and the UACR (*p* < 0.001) were significantly higher whereas serum c-peptide (*p* < 0.001), HOMA-β (*p* < 0.001) and uric acid (*P* = 0.011) levels were significantly lower in diabetic children than in the controls. Plasma KIM-1 (*P* = 0.007), urinary NGAL/Cr (*p* < 0.001) and urinary KIM-1/Cr (*P* = 0.005) were significantly higher in the diabetic children than in the controls. The UACR of diabetic children (11.57 mg/g (6.75–22.68 mg/g)) was within the NA range although it was significantly higher than in the controls (4.72 mg/g (3.49–8.81 mg/g)). The eGFR did not significantly differ between the two groups.

The baseline clinical and laboratory characteristics including plasma and urinary biomarker measurements in diabetic children with regard to the type of diabetes are presented in Table 2. Age (*p* < 0.001), body mass index (BMI) SDS (*p* < 0.001), serum protein (*p* < 0.001), AST (*P* = 0.038), ALT (*p* < 0.001), TG (*P* = 0.013), c-peptide (*p* < 0.001), HOMA-IR (*p* < 0.001), HOMA-β (*p* < 0.001) and uric acid (*P* = 0.002) were significantly higher whereas HDL-C (*p* < 0.001) levels were lower in the children with T2DM. However, plasma and urinary NGAL and KIM-1 showed no significant difference between the two groups.

### 3.2. Renal Injury Markers in Association with the MA Status

In the 54 diabetic children, the number of children classified as having normo-NA, high-NA and MA was 25 (46.3%), 19 (35.2%) and 10 (18.5%), respectively. The plasma KIM-1 level significantly differed among these groups (*P* = 0.008) (Figure 1A–D). Otherwise, no significant differences in the plasma NGAL, urinary NGAL/Cr and urinary KIM-1/Cr were observed among the three groups.

### 3.3. Association of Plasma KIM-1 with the Clinical and Laboratory Data

Plasma KIM-1 showed a significant correlation with age at the time of DM diagnosis, BMI SDS, serum glucose, protein, AST, ALT, TC, HDL-C, LDL-C, HbA1c, c-peptide, uric acid, HOMA-IR, UPCR and the UACR (Table 3).

The age at DM diagnosis, BMI SDS, serum glucose, AST, ALT, HDL-C, HbA1c and uric acid levels were identified as significant risk factors associated with increased plasma KIM-1 in diabetic children by univariate analyses (Table 4). Subsequently, multivariate analyses were performed after adjusting all significant variables identified in the univariate analyses. Finally, HbA1c remained significant as an independent risk factor of increased plasma KIM-1.

### 3.4. The HbA1c Cutoff Predicting a High Plasma KIM-1 Level

The cutoff point of increased plasma KIM-1 in our cohort was 23.10 pg/mL, which was approximated by the upper tertile (67th percentiles) value of the plasma KIM-1 concentration from a recent publication conducted in adults with non-proteinuric T1DM [7]. To evaluate the performance of plasma KIM-1 and determine the optimal glycemic target, a ROC curve was generated based on the cutoff of plasma KIM-1. The AUC was calculated as 0.77 (*p* < 0.001) and the optimal cutoff value of HbA1c was 6.75% (sensitivity, 0.68; specificity, 0.66) (Figure 2).

## 4. Discussions

This study aimed to evaluate the plasma and urinary NGAL and KIM-1 levels in diabetic children and determine their roles in predicting renal injury. Despite a lack of evidence and age-dependent changes in the reference interval for plasma and urinary NGAL and KIM-1, many publications have highlighted them as promising biomarkers reflecting early renal tubular injury in patients with CKD including DN [18,26,27]. The results of the present study were consistent with those of such previous studies. Plasma KIM-1, urinary KIM-1 and urinary NGAL levels were significantly elevated in diabetic children compared with the controls. The plasma KIM-1 levels in particular were higher in the high-NA state than in the normo-NA state, showing a gradual increase as albuminuria advanced although the levels between a few of the subjects with normo-NA and high-NA overlapped. These findings suggested that plasma KIM-1 could be a potential surrogate reflecting renal injury in diabetic children in the pre-albuminuric state. Additionally, our study focused on the role of plasma KIM-1 in normoalbuminuric diabetic children while other studies have mainly discussed the clinical implication of urinary KIM-1 in the prevention of DN.

KIM-1 is released into the interstitium and then the bloodstream due to an increased epithelial permeability and the loss of tubular cell polarity after tubular injury and its elevation indicates the risk of ESRD in diabetic patients independent of albuminuria [7,27]. In a previous study conducted in children with CKD, the plasma KIM-1 levels in the highest quartile were associated with a four-fold higher risk of CKD progression compared with the lowest quartile and were also associated with CKD progression of a non-glomerular origin [28]. Another study reported that plasma KIM-1 was one of the strongest renal injury markers associated with eGFR and increased before the UACR, suggesting that plasma KIM-1 is a more sensitive indicator of early renal damage [29]. Nevertheless, no pediatric reference data for normal plasma KIM-1 are available to differentiate unhealthy from healthy individuals.

Although the UACR was not identified as an independent risk factor for increased plasma KIM-1 in this study, the plasma KIM-1 levels were significantly higher in diabetic children with high-NA than in those with normo-NA. In addition, the plasma KIM-1 levels were highest in those with MA. MA is a risk factor not only for DN but also for other vascular complications such as diabetic retinopathy and cardiovascular diseases [30]. Recent studies have suggested lower UACR cutoffs to detect NA and MA for the earlier detection of CKD and DN [9]. Furthermore, many studies have reported individual MA cutoffs for different diabetic conditions such as 10.00 mg/g Cr for diabetic retinopathy and 19.25 mg/g Cr for hypertension [6]. Likewise, our study demonstrated that lowering the UACR threshold (to <10.00 mg/g Cr) could lead to the early detection of DN development. In addition, the present results suggested that plasma KIM-1 may help identify the risk of renal injury in diabetic children as a complementary marker to standard MA especially for those with mildly increased albuminuria.

HbA1c is known as the most significant indicator of long-term glycemic control and future prevention of DN. Early and routine screening of MA based on the UACR assessment in diabetic children is a critical step; however, a normal UACR does not preclude the absence of renal injury [31]. Our findings identified HbA1c as an independent risk factor for the elevation of plasma KIM-1. Our findings also suggested an HbA1c cutoff of 6.75% for increasing plasma KIM-1 levels, which implied that a universal target of HbA1c below 7.0% may not be sufficient particularly in association with renal injury [21,32]. Therefore, the risk of progression to DN may be reduced by focusing on an optimal glycemic control rather than solely depending on the albuminuric state. Lowering the HbA1a level can be achieved by intensive treatment and glucose-lowering efforts by patients. This is particularly crucial in young children with DM who require a greater intervention to prevent long-term microvascular complications.

A few limitations of this study need to be addressed. First, this was not an age-matched case-controlled study and was performed on a small sample size within a cross-sectional design. Although a significant difference in plasma KIM-1 levels was observed based on the albuminuric state, the number of children with MA was insufficient to accurately determine the association of the MA state with plasma and urinary NGAL and KIM-1 levels. Secondly, the BMI SDS of the controls was 2.22 (0.78–2.92), which classified them as obese children; selection bias may have occurred in this respect. Obesity could have affected the NGAL and KIM-1 levels and, therefore, the outcome and interpretation might have been altered. Thirdly, blood pressure could have affected NGAL and KIM-1 levels; however, statistical analyses were not performed because we were unable to collect blood pressure data from all of the study participants. Lastly, the AUC of the ROC curve for optimal HbA1c was significant but relatively low-powered with a moderate sensitivity and specificity. Nevertheless, the most important strength of the present study was that we highlighted the significance of plasma KIM-1 for the early detection of DN in normoalbuminuric diabetic children and concluded that a tight glycemic control was the most important action against the progression to incipient DN by suggesting a potential HbA1c target. Additionally, the data on plasma KIM-1 levels of non-diabetic children could be valuable for future reference considering the lack of evidence.

## 5. Conclusions

In conclusion, plasma KIM-1 was higher in diabetic compared with non-diabetic children and tended to gradually increase in diabetic children depending on the albuminuric state. For diabetic children and adolescents with mildly increased albuminuria, plasma KIM-1 complementary to standard MA may help increase the yield of renal injury detection. HbA1c higher than 6.75% may be an independent risk factor for increased plasma KIM-1 levels. Above all, the patient’s effort in improving the glycemic control and lowering HbA1c should be prioritized to minimize long-term vascular complications. Larger prospective studies are required to further investigate the determinants of KIM-1 elevation in early DN for the earlier detection of childhood renal injury.

## Figures and Tables

**Figure 1 children-08-00417-f001:**
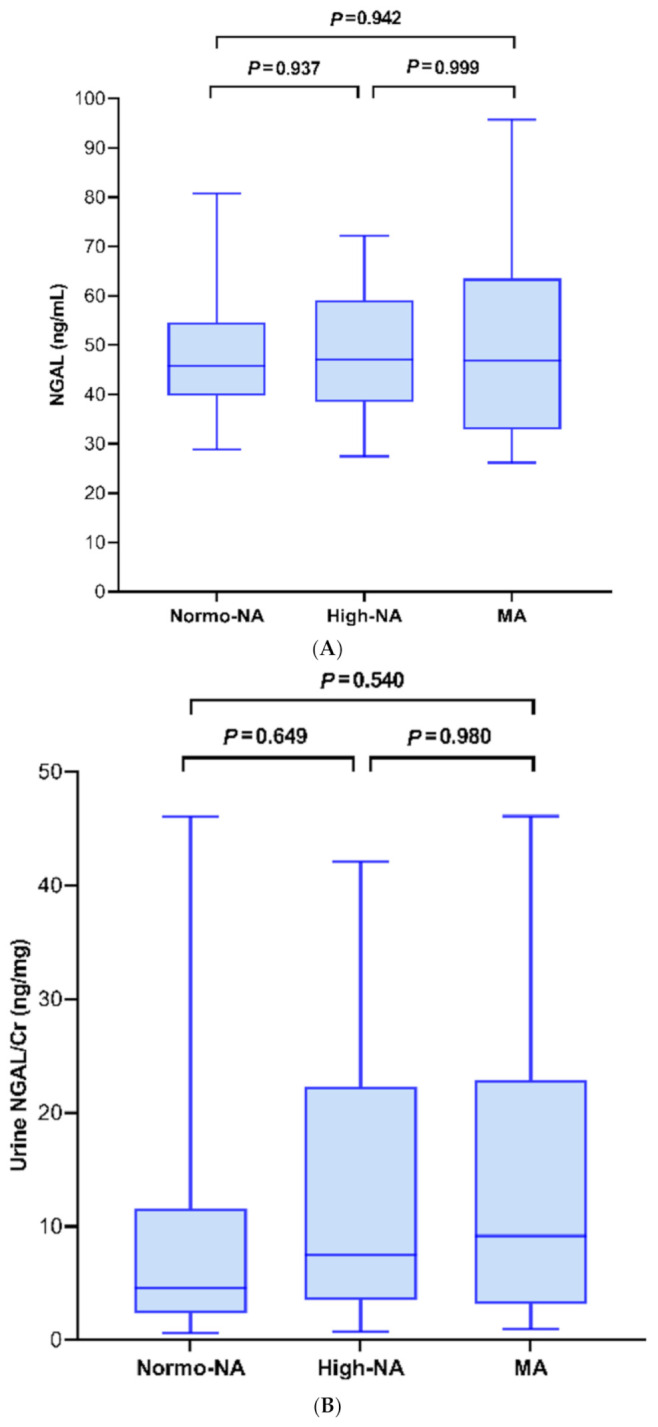
Diabetic children were subcategorized into three groups with regard to the albumin creatinine ratio and box plots were drawn to compare: (**A**) serum neutrophil gelatinase-associated lipocalin (NGAL), (**B**) urine NGAL/g creatinine, (**C**) serum kidney injury molecule-1 (KIM-1) and (**D**) urine KIM-1/g creatinine. The boxes represent the interquartile range. The lines inside the boxes represent the median value. The whisker represents the lowest and highest observations, respectively.

**Figure 2 children-08-00417-f002:**
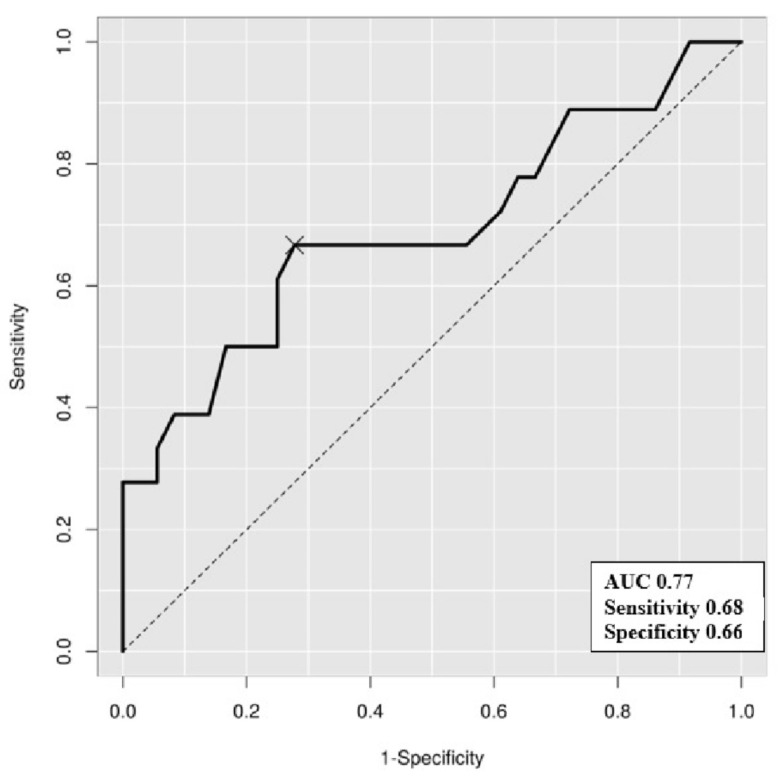
Receiver operating characteristic (ROC) curve of glycosylated hemoglobin and plasma kidney injury molecule-1 greater than 23.10 pg/mL. The area under the curve (AUC) was 0.77 and the optimal cutoff point (sensitivity, specificity) was 6.75% (0.68, 0.66).

**Table 1 children-08-00417-t001:** Clinical characteristics and plasma and urine biomarker measurements of the study subjects.

	Total (*n* = 98)		
	Control (*n* = 44)	Diabetic Children (*n* = 54)	*p*
Gender, male/female, *n*	27/17	28/26	0.415
Age (years)	9.92 (8.67–11.58)	15.83 (13.46–17.69)	<0.001
BMI SDS	2.22 (0.78–2.92)	1.64 (−0.65–3.08)	0.389
Hemoglobin (g/dL)	13.6 (13.1–14.2)	14.1 (13.28–15.60)	0.059
CRP (mg/dL)	0.07 (0.03–0.14)	0.30 (0.08–1.46)	<0.001
Glucose (mg/dL)	94.00 (90.75–97.25)	118.00 (104.25–185.25)	<0.001
Cr (mg/dL)	0.47 (0.43–0.55)	0.56 (0.48–0.70)	<0.001
eGFR (mL/min/1.73 m^2^)	121.18 (107.21–129.78)	118.15 (99.79–133.13)	0.337
Protein (mg/dL)	7.40 (7.10–7.70)	7.30 (7.10–7.60)	0.435
Albumin (g/dL)	4.70 (4.60–4.90)	4.70 (4.50–4.90)	0.311
AST (U/L)	22.50 (18.75–28.25)	18.50 (13.75–34.75)	0.080
ALT (U/L)	15 (11.75–22.50)	14.00 (10.00–56.25)	0.689
TC (mg/dL)	173.50 (155.75–203.75)	179.00 (156.75–202.50)	0.898
TG (mg/dL)	97.00 (58.75–133.50)	110.50 (58.00–171.75)	0.530
HDL-C (mg/dL)	50.00 (44.00–60.25)	52.00 (44.00–633.25)	0.825
LDL-C (mg/dL)	106.00 (90.75–127.00)	106.10 (78.75–124.00)	0.328
HbA1c (%)	5.40 (5.20–5.60)	8.20 (6.70–9.70)	<0.001
C-peptide (ng/mL)	2.45 (1.83–3.42)	0.78 (0.11–2.88)	<0.001
HOMA-IR	3.87 (2.26–5.72)	2.02 (0.76–6.91)	0.070
HOMA-β	200.35 (136.19–292.07)	40.49 (14.00–111.01)	<0.001
Uric acid (mg/dL)	5.25 (4.30–6.85)	4.85 (4.08–5.60)	0.011
UPCR (mg/g Cr)	0.09 (0.08–0.12)	0.09 (0.06–0.13)	0.749
UACR (mg/g Cr)	4.72 (3.49–8.81)	11.57 (6.75–22.68)	<0.001
NGAL (ng/mL)	46.42 (37.63–53.71)	45.96 (38.60–55.95)	0.828
Urine NGAL/Cr (ng/mg Cr)	2.78 (1.23–5.83)	5.91 (2.83–14.70)	<0.001
KIM-1 (pg/mL)	11.25 (9.06–14.62)	16.89 (9.10–29.27)	0.007
Urine KIM-1/Cr (ng/mg Cr)	0.61 (0.41–0.81)	1.01 (0.49–1.64)	0.005

Data presented as a median (interquartile range (IQR) 25–75%). Laboratory measurements are serum values unless otherwise noted. ALT, alanine aminotransferase; AST, aspartate aminotransferase; BMI, body mass index; Cr, creatinine; CRP, C-reactive protein; eGFR, estimated glomerular filtration rate; HbA1c, glycosylated hemoglobin; HDL-C, high-density lipoprotein cholesterol; HOMA-IR, homeostatic model assessment of insulin resistance; HOMA-β, homeostatic model assessment of beta cell function; KIM-1, kidney injury molecule-1; LDL-C, low-density lipoprotein cholesterol; NGAL, neutrophil gelatinase-associated lipocalin; SDS, standard deviation score; TC, total cholesterol; TG, triglycerides; UACR, urine albumin/creatinine ratio; UPCR, urine protein/creatinine ratio.

**Table 2 children-08-00417-t002:** Clinical characteristics and plasma and urine biomarker measurements of diabetic children with regard to the type of diabetes.

	Total (*n* = 98)		
	T1DM (*n* = 31)	T2DM (*n* = 23)	*p*
Gender, male/female, *n*	18/13	10/13	0.409
Age (years)	14.67 (10.08–16.42)	17.58 (15.75–18.25)	<0.001
Duration of known DM (years)	2.00 (0.75–4.50)	1.33 (1.17–3.42)	0.353
BMI SDS	0.15 (−1.16–1.59)	3.19 (1.99–4.66)	<0.001
Hemoglobin (g/dL)	13.70 (13.10–15.50)	14.30 (13.60–15.60)	0.244
CRP (mg/dL)	0.30 (0.03–0.73)	0.87 (0.12–2.60)	0.062
Glucose (mg/dL)	113.00 (101.00–161.00)	122.00 (112.00–219.00)	0.124
Cr (mg/dL)	0.53 (0.45–0.70)	0.60 (0.54–0.72)	0.050
eGFR (mL/min/1.73 m^2^)	123.90 (102.06–138.79)	116.60 (94.36–131.70)	0.340
Protein (mg/dL)	7.20 (7.00–7.40)	7.50 (7.30–7.80)	<0.001
Albumin (g/dL)	4.60 (4.50–4.90)	4.70 (4.50–4.90)	0.487
AST (U/L)	17.00 (14.00–21.00)	34.00 (13.00–59.00)	0.038
ALT (U/L)	12.00 (9.00–15.00)	57.00 (12.00–103.00)	<0.001
TC (mg/dL)	170.00 (149.00–204.00)	181.00 (159.00–202.00)	0.903
TG (mg/dL)	79.00 (50.00–156.00)	127.00 (99.00–230.00)	0.013
HDL-C (mg/dL)	59.00 (52.00–71.00)	44.00 (39.00–49.00)	<0.001
LDL-C (mg/dL)	90.40 (74.00–124.00)	116.60 (90.00–123.20)	0.128
HbA1c (%)	8.50 (6.80–9.60)	7.90 (6.50–10.30)	0.759
C-peptide (ng/mL)	0.13 (0.02–0.66)	3.26 (1.96–4.61)	<0.001
HOMA-IR	0.95 (0.38–1.82)	6.59 (4.27–12.84)	<0.001
HOMA-β	22.76 (9.53–42.45)	107.92 (60–04–209.39)	<0.001
Uric acid (mg/dL)	4.40 (3.80–5.30)	5.40 (4.70–6.40)	0.002
UPCR (mg/g Cr)	0.10 (0.08–0.16)	0.08 (0.06–0.11)	0.098
UACR (mg/g Cr)	9.43 (6.51–20.99)	14.98 (6.99–25.32)	0.535
NGAL (ng/mL)	41.77 (37.05–49.87)	51.11 (45.76–62.71)	0.051
Urine NGAL/Cr (ng/mg Cr)	4.68 (3.16–23.72)	6.61 (2.05–13.67)	0.588
KIM-1 (pg/mL)	14.84 (8.00–20.67)	22.64 (12.36–33.65)	0.051
Urine KIM-1/Cr (ng/mg Cr)	1.00 (0.55–1.68)	0.97 (0.39–1.60)	0.601

Data presented as a median (interquartile range (IQR) 25–75%). Laboratory measurements are serum values unless otherwise noted. ALT, alanine aminotransferase; AST, aspartate aminotransferase; BMI, body mass index; Cr, creatinine; CRP, C-reactive protein; DM, diabetes mellitus; eGFR, estimated glomerular filtration rate; HbA1c, glycosylated hemoglobin; HDL-C, high-density lipoprotein cholesterol; HOMA-IR, homeostatic model assessment of insulin resistance; HOMA-β, homeostatic model assessment of beta cell function; KIM-1, kidney injury molecule-1; LDL-C, low-density lipoprotein cholesterol; NGAL, neutrophil gelatinase-associated lipocalin; SDS, standard deviation score; T1DM, type 1 diabetes mellitus; T2DM, type 2 diabetes mellitus; TC, total cholesterol; TG, triglycerides; UACR, urine albumin/creatinine ratio; UPCR, urine protein/creatinine ratio.

**Table 3 children-08-00417-t003:** Spearman’s rank correlation of plasma kidney injury molecule-1 with clinical and laboratory characteristics of diabetic children.

	KIM-1	
	Spearman’s ρ	*P*
Age at DM diagnosis	0.356	0.008
Duration of known DM	−0.152	0.272
Age	0.163	0.238
BMI SDS	0.643	<0.001
Hemoglobin	0.140	0.314
CRP	0.243	0.077
Glucose	0.380	0.005
Cr	−0.078	0.577
eGFR	0.141	0.309
Protein	0.324	0.017
Albumin	0.003	0.980
AST	0.431	0.001
ALT	0.542	<0.001
TC	0.466	<0.001
TG	0.639	0.079
HDL-C	−0.393	0.003
LDL-C	0.360	0.007
HbA1c	0.368	0.006
C-peptide	0.293	0.032
HOMA-IR	0.377	0.005
HOMA-β	0.122	0.378
Uric acid	0.360	0.008
UPCR	0.316	0.020
UACR	0.398	0.003

Laboratory measurements are serum values unless otherwise noted. ALT, alanine aminotransferase; AST, aspartate aminotransferase; BMI, body mass index; Cr, creatinine; CRP, C-reactive protein; DM, diabetes mellitus; eGFR, estimated glomerular filtration rate; HbA1c, glycosylated hemoglobin; HDL-C, high-density lipoprotein cholesterol; HOMA-IR, homeostatic model assessment of insulin resistance; HOMA-β, homeostatic model assessment of beta cell function; KIM-1, kidney injury molecule-1; LDL-C, low-density lipoprotein cholesterol; SDS, standard deviation score; TC, total cholesterol; TG, triglycerides; UACR, urine albumin/creatinine ratio; UPCR, urine protein/creatinine ratio.

**Table 4 children-08-00417-t004:** Univariate and multivariate regression analyses of factors associated with increased plasma KIM-1.

Risk Factors	Univariate			Multivariate		
	OR (95% CI)	SE	*P*	OR (95% CI)	SE	*p*
DM type	7.89 (−2.77–18.6)	5.31	0.143			
Age at DM diagnosis	1.98 (0.47–3.49)	0.75	0.011	1.16 (–0.74–3.06)	0.94	0.226
DM duration	−0.93 (−2.98–1.12)	1.02	0.367			
BMI SDS	4.63 (2.51–6.75)	1.06	<0.001	1.72 (–1.62–5.06)	1.66	0.305
Hb	1.21 (−2.30–4.73)	1.75	0.492			
CRP	0.25 (−1.13–1.63)	0.69	0.721			
Glucose	0.09 (<0.01–0.17)	0.04	0.039	−0.03 (−0.13–0.06)	0.05	0.460
Cr	18.6 (−14.12–51.30)	16.30	0.259			
eGFR	−0.07 (−0.30–0.16)	0.11	0.542			
Protein	7.02 (−3.17–17.20)	5.08	0.173			
Albumin	−2.33 (−16.60–12.00)	7.13	0.745			
AST	0.45 (0.24–0.67)	0.11	<0.001	0.25 (−0.37–0.87)	0.31	0.415
ALT	0.24 (0.14–0.35)	0.05	<0.001	0.02 (−0.33–0.37)	0.17	0.906
TC	0.19 (0.02–0.35)	0.08	0.090			
TG	0.12 (0.07–0.17)	0.02	0.061			
HDL-C	−0.40 (−0.77–0.03)	0.19	0.035	0.08 (−0.34–0.51)	0.21	0.700
LDL-C	0.16 (−0.03–0.34)	0.09	0.102			
HbA1c	2.99 (0.60–5.38)	1.19	0.015	2.96 (0.35–5.56)	1.29	0.027
C-peptide	2.73 (−0.33–5.80)	1.53	0.079			
HOMA-IR	0.37 (−0.11–0.85)	0.24	0.127			
HOMA-β	−0.01 (−0.06–0.05)	0.03	0.813			
Uric acid	5.21 (0.86–9.56)	2.17	0.020	1.38 (−4.45–7.22)	2.89	0.636
UPCR	5.16 (−25.40–35.80)	15.25	0.736			
UACR	0.03 (−0.05–0.09)	0.04	0.461			

Laboratory measurements are serum values unless otherwise noted. ALT, alanine aminotransferase; AST, aspartate aminotransferase; BMI, body mass index; Cr, creatinine; CRP, C-reactive protein; eGFR, estimated glomerular filtration rate; HbA1c, glycosylated hemoglobin; HDL-C, high-density lipoprotein cholesterol; HOMA-IR, homeostatic model assessment of insulin resistance; HOMA-β, homeostatic model assessment of beta cell function; KIM-1, kidney injury molecule-1; LDL-C, low-density lipoprotein cholesterol; SDS, standard deviation score; TC, total cholesterol; TG, triglycerides; UACR, urine albumin/creatinine ratio; UPCR, urine protein/creatinine ratio.

## Data Availability

The raw data supporting the conclusions of this article will be made available by the authors without undue reservation.

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
