# Peer review of "Poor Glycemic Control Can Increase the Plasma Kidney Injury Molecule-1 Concentration in Normoalbuminuric Children and Adolescents with Diabetes Mellitus"

_children, 2021, doi:10.3390/children8050417_

Round 1
Reviewer 1 Report
The authors reported that plasma KIM-1 could be a useful indicator of renal injury in diabetic children. It is so interesting theme, but I feel there are several issues to be solved.
- According to Figure 1, plasma KIM-1 levels in Normo-NA and High-NA groups were largely merged. Thus, I feel that plasma KIM-1 level could not be a biomarker to distinguish Normo-NA and High-NA groups.
- I do not agree to analyze type 1 and type 2 diabetic patients simultaneously.
- In Table 2, how did you choose the multivariate model? You should describe the reason in the text.
- You should mention about the medication including insulin therapy of the diabetic subjects.
Author Response
Thank you for your insightful and thorough comments. The attached is our response. Thank you again.

Reviewer 2 Report
In this article, the authors have described that poor glycemic control can increase plasma kidney injury molecule-1 concentration in normoalbuminuric children and adolescents with diabetes mellitus. I have no major concerns and this article is well written. I have the following minor comments.
- Did authors analyze the blood pressure between the different cohorts.
- This sentence in the abstract "Plasma KIM-1 could be a useful indicator of renal injury in normoalbuminuric diabetic children and, and HbA1c<7.0% might be an independent risk factor for increased plasma KIM-1 levels" lacks clarity. I would suggest to rephrase as follows "our findings also suggested an HbA1c cutoff of 6.75% corelated with increased plasma KIM-1 levels" This also applies in the conclusion section as well.
Author Response

(The authors gave the same response as above.)
